# Intravascular flow stimulates PKD2 (polycystin-2) channels in endothelial cells to reduce blood pressure

Charles E MacKay, M Dennis Leo, Carlos Fernández-Peña, Raquibul Hasan, Wen Yin, Alejandro Mata-Daboin, Simon Bulley, Jesse Gammons, Salvatore Mancarella, Jonathan H Jaggar*

Department of Physiology University of Tennessee Health Science Center Memphis, Memphis, United States

**Abstract** PKD2 (polycystin-2, TRPP1), a TRP polycystin channel, is expressed in endothelial cells (ECs), but its physiological functions in this cell type are unclear. Here, we generated inducible, EC-specific *Pkd2* knockout mice to examine vascular functions of PKD2. Data show that a broad range of intravascular flow rates stimulate EC PKD2 channels, producing vasodilation. Flow-mediated PKD2 channel activation leads to calcium influx that activates SK/IK channels and eNOS serine 1176 phosphorylation in ECs. These signaling mechanisms produce arterial hyperpolarization and vasodilation. In contrast, EC PKD2 channels do not contribute to acetylcholine-induced vasodilation, suggesting stimulus-specific function. EC-specific PKD2 knockout elevated blood pressure in mice without altering cardiac function or kidney anatomy. These data demonstrate that flow stimulates PKD2 channels in ECs, leading to SK/IK channel and eNOS activation, hyperpolarization, vasodilation and a reduction in systemic blood pressure. Thus, PKD2 channels are a major component of functional flow sensing in the vasculature.

*For correspondence:
jjaggar@uthsc.edu

Competing interests: The authors declare that no competing interests exist.

## Introduction

Endothelial cells line the lumen of all blood vessels and regulate multiple functions, including contractility. A wide variety of different stimuli act through endothelial cells to control arterial contractility, including receptor ligands, such as acetylcholine (ACh), and mechanical force, including intravascular flow. Mechanisms by which endothelial cells regulate arterial contractility include the production of diffusible substances, such as nitric oxide (NO) and hydrogen sulfide, and the release of endothelium-derived hyperpolarizing factors, including potassium ($K^+$) (*Vane, 1994*; *Edwards et al., 1998*; *Leffler et al., 2006*). Due to the presence of myoendothelial gap junctions, endothelial cells can also directly control smooth muscle cell membrane potential to regulate arterial contractility (*Garland et al., 2011*). Less well defined are signaling mechanisms by which physiological stimuli activate these processes in endothelial cells to produce vasodilation. In particular, the regulatory mechanisms, physiological functions and in vivo significance of many ion channels that are expressed in endothelial cells are poorly understood.

Endothelial cells express several different families of ion channels, including multiple transient receptor potential (TRP), small-conductance $Ca^{2+}$-activated $K^+$ (SK3, $K_{Ca}2.3$) and intermediate-conductance $Ca^{2+}$-activated $K^+$ (IK, $K_{Ca}3.1$) proteins (*Jackson, 2016*). TRP channels are a family of ~28 proteins that are subdivided into six different classes, including polycystin (TRPP), melastatin (TRPM), ankyrin (TRPA), canonical (TRPC) and vanilloid (TRPV) (*Earley and Brayden, 2015*). Studies performed using whole arteries and veins, which contain multiple different cell types, and cultured and non-cultured cells have proposed that approximately twenty different TRP channels may be expressed in endothelial cells (*Sullivan and Earley, 2013*; *Bulley et al., 2018*). A significant body of

work indicates that TRPV4 channels present in endothelial cells regulate the contractility of vasculature, including resistance-size arteries (*Earley et al., 2009*; *Sonkusare et al., 2012*; *Zhang et al., 2009*; *Köhler et al., 2006*; *Marrelli et al., 2007*). Evidence also suggests that endothelial cell TRPA1, TRPC3, TRPC4, TRPV1 and TRPV3 channels modulate vascular contractility (*Liu et al., 2006*; *Gao et al., 2012*; *Freichel et al., 2001*; *Yang et al., 2010*; *Bratz et al., 2008*; *Earley et al., 2010*; *Sullivan et al., 2015*). In many of these previous studies, TRP channel expression and or function was reported in endothelial cells of ex vivo vasculature that does not control systemic blood pressure, including conduit vessels, cerebral arteries, mammary arteries and umbilical vein (*Earley and Brayden, 2015*; *Gao et al., 2012*). Physiological functions of many TRP channels that are proposed to be expressed in endothelial cells are poorly understood, particularly in small resistance-size arteries that regulate regional organ blood flow and systemic blood pressure.

PKD2, which is also termed Transient Receptor Potential Polycystin 1 (TRPP1), PC-2 and polycystin-2, is encoded by the *Pkd2* gene (*Mochizuki et al., 1996*). PKD2 contains six transmembrane domains, cytoplasmic N and C termini and a characteristic extracellular polycystin domain (*Shen et al., 2016*). PKD2 protein is expressed in a wide variety of different cell types, including endothelium, arterial smooth muscle, renal epithelia, cardiac myocytes and neurons, (*Bulley et al., 2018*; *Semmo et al., 2014*). Mutations in *Pkd2* lead to Autosomal Dominant Polycystic Kidney Disease (ADPKD), the most prevalent monogenic human disease worldwide (*Torres et al., 2007*). ADPKD is typically characterized by the growth of renal cysts, although a significant proportion of patients develop hypertension prior to kidney dysfunction, suggesting PKD2 channels perform physiological functions in vascular wall cell types (*Torres et al., 2007*; *Valero et al., 1999*; *Martinez-Vea et al., 2004*). We have previously shown that intravascular pressure and $\alpha_1$-adrenoceptors activate PKD2 channels in arterial smooth muscle cells of different organs, leading to depolarization, vasoconstriction and an increase in systemic blood pressure (*Bulley et al., 2018*). In contrast, regulatory mechanisms and physiological functions of PKD2 channels in endothelial cells are unclear.

Here, we developed an inducible, cell-specific, knockout mouse model to study physiological functions of PKD2 channels in endothelial cells. We show that intravascular flow stimulates PKD2 channels in endothelial cells and that this mechanism is a major contributor to flow-mediated vasodilation over a broad shear stress range. In contrast, PKD2 channels do not contribute to ACh-induced dilation, suggesting stimulus-specific function. Flow-mediated PKD2 channel activation leads to $Ca^{2+}$ influx, which activates SK and IK channels, and stimulates eNOS. These mechanisms induce arterial hyperpolarization, vasodilation and a reduction in blood pressure. Thus, PKD2 channels are a major contributor to functional flow-sensing in endothelial cells.

## Results

### Generation of tamoxifen-inducible, endothelial cell-specific PKD2 knockout mice

Mice with *loxP* sites flanking exons 11 and 13 (*Pkd2^fl/fl^*) of the *Pkd2* gene were crossed with tamoxifen-inducible, endothelial cell-specific Cre (*Cdh5*-creERT2) mice, producing a *Pkd2^fl/fl^:Cdh5*-creERT2 line. Genomic PCR confirmed that tamoxifen stimulated *Pkd2* recombination in mesenteric arteries of *Pkd2^fl/fl^:Cdh5*-creERT2 mice, but not in arteries of *Pkd2^fl/fl^* mice (*Figure 1—figure supplement 1*). Genomic PCR also amplified an identical product in tamoxifen-treated *Pkd2^fl/fl^* and *Pkd2^fl/fl^:Cdh5*-creERT2 mouse arteries due to *Pkd2* in cells such as smooth muscle, where DNA would not undergo recombination (*Figure 1—figure supplement 1*; *Bulley et al., 2018*).

Western blotting was performed to quantify proteins in lysate collected from second- through fifth-order mesenteric artery branches. PKD2 protein in mesenteric arteries of tamoxifen-treated *Pkd2^fl/fl^:Cdh5*-creERT2 mice was ~ 67.2% of that in tamoxifen-treated *Pkd2^fl/fl^* controls (*Figure 1A, B*). This reduction in total arterial protein is expected given that smooth muscle cells, which also express PKD2, are far more abundant than endothelial cells in vessels of this size (*Bulley et al., 2018*). These data are also consistent with our previous observation that smooth muscle-specific PKD2 knockout reduced total mesenteric arterial wall PKD2 protein by ~ 75% (*Bulley et al., 2018*). In contrast, SK3, IK, TRPV4, Piezo1, GPR68 and PKD1 (polycystin-1, PC-1), which can form a complex with PKD2 (*Qian et al., 1997*; *Tsiokas et al., 1997*), were similar in arteries of both genotypes (*Figure 1a and b*). Immunofluorescence demonstrated that PKD2 protein was present in endothelial

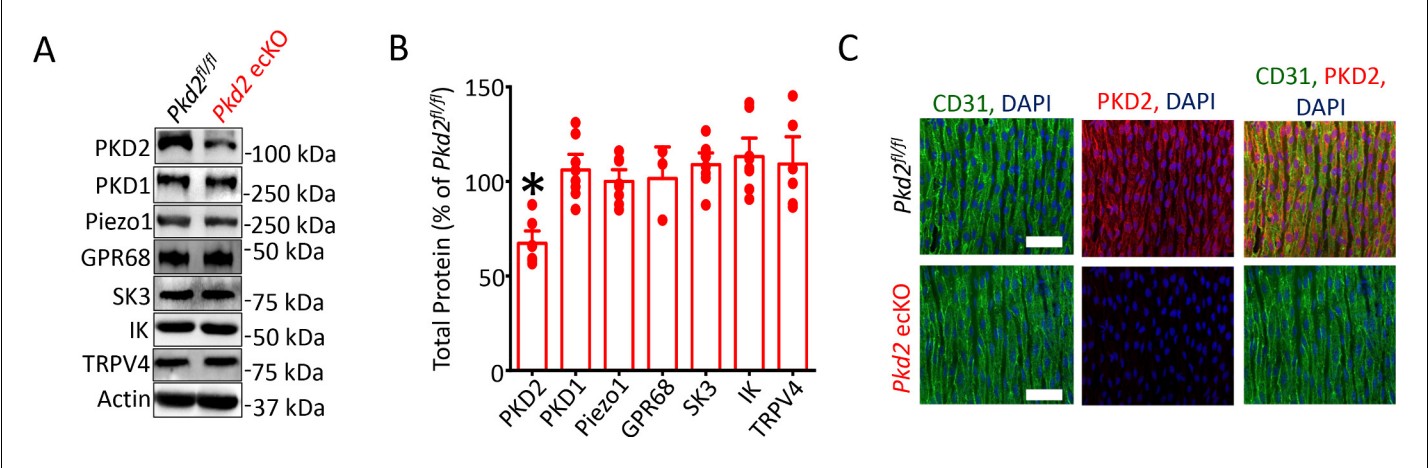

**Figure 1.** Generation and validation of *Pkd2* ecKO mice. (**A**) Representative Western blots illustrating the effect of tamoxifen-treatment of *Pkd2*flfl and *Pkd2*flfl: *Cdh5*(PAC)-creERT2 mice on PKD2, PKD1, Piezo1, GPR68, eNOS, SK3, IK and TPRV4, proteins in mesenteric arteries. (**B**) Mean data for proteins in mesenteric arteries of tamoxifen-treated *Pkd2*flfl: *Cdh5*(PAC)-creERT2 mice when compared to those in tamoxifen-treated *Pkd2*flfl mice. n = 3–8. * indicates $p < 0.05$ versus *Pkd2*flfl. (**C**) *En-face* immunofluorescence imaging illustrating that PKD2 protein (Alexa Fluor 555) is abolished in endothelial cells of mesenteric arteries in tamoxifen-treated *Pkd2*flfl: *Cdh5*(PAC)-creERT2 mice (representative of 6 mesenteric arteries). CD31 (Alexa Fluor 488) and DAPI are also shown. Scale bars = 50 μm.

The online version of this article includes the following figure supplement(s) for figure 1:

**Figure supplement 1.** Genotyping of mouse lines.

cells of intact arteries from tamoxifen-treated *Pkd2*flfl mice, but absent in endothelial cells of tamoxifen-treated *Pkd2*flfl:*Cdh5*-creERT2 mice (*Figure 1C*). These results indicate that PKD2 is expressed in endothelial cells and suggest that tamoxifen treatment of *Pkd2*flfl:*Cdh5*-creERT2 mice abolishes PKD2 protein. Tamoxifen-treated *Pkd2*flfl:*Cdh5*-creERT2 mice will thus be referred to as *Pkd2* ecKO mice. Tamoxifen-treated *Pkd2*flfl mice were used as controls in all experiments.

## Endothelial cell PKD2 channels contribute to flow-, but not ACh-, mediated vasodilation

To investigate physiological functions of endothelial cell PKD2 channels, diameter responses to vasoactive stimuli were measured in pressurized (80 mmHg) third-order mesenteric arteries of *Pkd2*flfl and *Pkd2* ecKO mice. Vasodilation to ACh, a muscarinic receptor agonist, was similar in control and *Pkd2* ecKO arteries, suggesting that endothelial cell PKD2 channels do not contribute to this response (*Figure 2A and C*). Repetitive intravascular flow (15 dyn/cm²) stimuli produced sustained, reproducible and fully reversible vasodilation in pressurized (80 mmHg) mesenteric arteries (*Figure 2—figure supplement 2A–D*). In pressurized *Pkd2* ecKO arteries, mean vasodilation to single on-off flow stimuli were ~35.1% of those in *Pkd2*flfl arteries (*Figure 2B and C*). Endothelial cell-denudation abolished vasodilation to both flow and ACh (*Figure 2—figure supplement 1A and B*). In contrast, endothelial denudation did not alter dilation to sodium nitroprusside, a NO donor, indicating that smooth muscle function was not altered by this procedure (*Figure 2—figure supplement 1A and B*). To determine the range over which endothelial cell PKD2 channels function, we measured vasoregulation to flow rates that produced shear stress between 3 and 35 dyn/cm². Cumulative increases in flow caused progressive dilation in *Pkd2*flfl arteries, with a maximum at 27 dyn/cm² (*Figure 2D,E*). Further increasing flow partially reduced this maximal vasodilatory response (*Figure 2D and E*). Flow stimulated less vasodilation in *Pkd2* ecKO arteries over the range studied (*Figure 2D,E*; *Figure 2—figure supplement 3*). Specifically, flow-mediated vasodilation was between ~ 45.5% and 60.1% of that in *Pkd2*flfl arteries, regardless of rate (*Figure 2D and E*, *Figure 2—figure supplement 3*). These data indicate that endothelial cell PKD2 channels function over a broad flow range to stimulate vasodilation in pressurized arteries.

Experiments were performed to examine the hypothesis that endothelial cell PKD2 channel knockout modifies smooth muscle cell contractility, thereby indirectly altering responses to flow. An

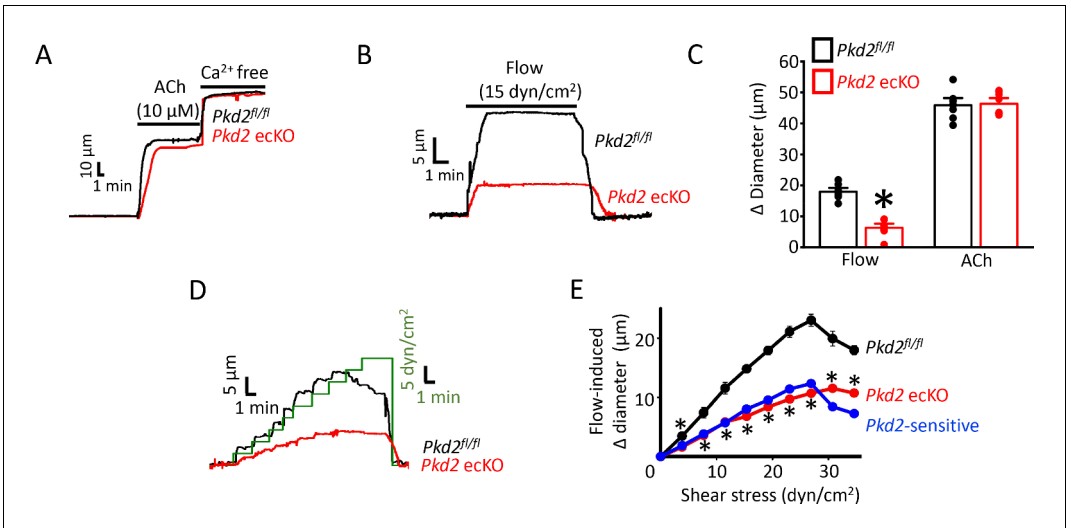

**Figure 2.** PKD2 channels contribute to intravascular flow-, but not ACh-, mediated vasodilation. (**A**) Original traces illustrating responses to ACh (10 μM) and $Ca^{2+}$-free solution (passive diameter) in pressurized (80 mmHg) mesenteric arteries from *Pkd2fl/fl* and *Pkd2* ecKO mice. (**B**) Original trace of flow-mediated dilation in pressurized (80 mmHg) mesenteric arteries from *Pkd2fl/fl* and *Pkd2* ecKO mice. (**C**) Mean diameter changes in response to flow (15 dyn/cm$^2$) or ACh (10 μM). *p<0.05 vs. *Pkd2fl/fl*. n = 8 for each. # p<0.05 vs. flow in the same genotype. (**D**) Original traces illustrating diameter responses to stepwise increases in intravascular flow in pressurized (80 mmHg) mesenteric arteries from *Pkd2fl/fl* and *Pkd2* ecKO mice. (**E**) Mean data. The *Pkd2*-sensitive component of flow-mediated vasodilation is illustrated in blue. *p<0.05 vs. *Pkd2fl/fl*. n = 5 for *Pkd2fl/fl*, n = 4 for *Pkd2* ecKO.

The online version of this article includes the following figure supplement(s) for figure 2:

**Figure supplement 1.** Endothelial denudation abolishes ACh-mediated vasodilation.

**Figure supplement 2.** Endothelial denudation abolishes flow-mediated dilation.

**Figure supplement 3.** Endothelial cell PKD2 knockout attenuates flow-mediated vasodilation over a broad shear stress range.

**Figure supplement 4.** Smooth muscle-specific vasoconstriction and passive diameter are unaltered in *Pkd2* ecKO arteries.

increase in extracellular potassium (60 mm $K^+$) or intravascular pressure (80 mmHg) similarly constricted arteries of *Pkd2fl/fl* and *Pkd2* ecKO mice, indicating that endothelial cell PKD2 channels or their knockout does not influence depolarization-induced vasoconstriction or myogenic tone, respectively (***Figure 2—figure supplement 4A–D***). Similarly, arterial passive diameter, determined by removal of extracellular $Ca^{2+}$ from the bath solution, was similar in *Pkd2fl/fl* and *Pkd2* ecKO arteries (***Figure 2—figure supplement 4E***). Thus, knockout of endothelial cell PKD2 channels does not modify smooth muscle cell function.

## Endothelial cell PKD2 channels contribute to flow-mediated arterial hyperpolarization

To investigate mechanisms by which endothelial cell PKD2 channels regulate contractility, membrane potential was measured in pressurized mesenteric arteries using glass microelectrodes. At 10 mmHg, the mean membrane potential of *Pkd2fl/fl* and *Pkd2* ecKO arteries were similar at ~ −62.1 and −61.1 mV, respectively (***Figure 3A,B***). Increasing intravascular pressure to 80 mmHg similarly depolarized *Pkd2fl/fl* and *Pkd2* ecKO arteries by ~ 19.3 and 17.7 mV, respectively (***Figure 3A,B***). Intravascular flow stimulated a mean hyperpolarization of ~ 11 mV in *Pkd2fl/fl* arteries (***Figure 3A,B***). In contrast, flow only hyperpolarized *Pkd2* ecKO arteries by ~ 3 mV, or ~ 25.5% of that in controls (***Figure 3A,B***). These data suggest that flow modulates PKD2 channels in endothelial cells, leading to arterial hyperpolarization and vasodilation.

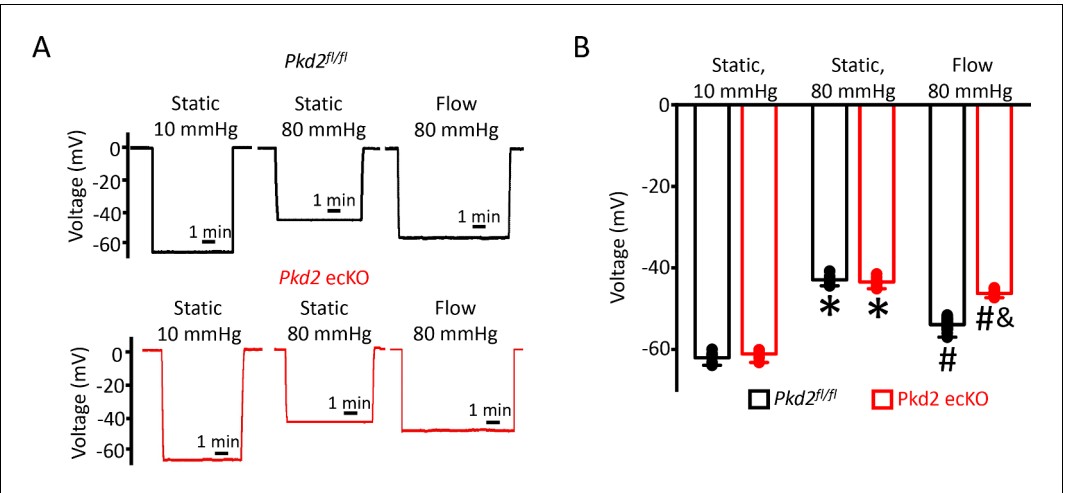

**Figure 3.** EC PKD2 channels contribute to flow-mediated arterial hyperpolarization. (A) Original membrane potential recordings obtained from microelectrode impalements in pressurized mesenteric arteries of $Pkd2^{fl/fl}$ and $Pkd2$ ecKO mice at static (10 and 80 mmHg) and with 80 mmHg and flow (15 dyn/cm$^2$). All three impalements in $Pkd2^{fl/fl}$ and $Pkd2$ ecKO were from the same two arteries. (B) Mean data ($Pkd2^{fl/fl}$: 10 mmHg, n = 8; 80 mmHg, n = 10; 80 mmHg + flow, n = 18; $Pkd2$ ecKO: 10 mmHg, n = 7; 80 mmHg, n = 12; 80 mmHg + flow, n = 16). *p<0.05 for 80 mmHg static versus 10 mmHg static in same genotype. # p<0.05 for 80 mmHg + flow versus 80 mmHg static in the same genotype. and indicates p<0.05 versus $Pkd2^{fl/fl}$ under the same condition.

## Flow activates a PKD2-mediated reduction in inward current in endothelial cells

The contribution of PKD2 channels to currents was investigated in mesenteric artery endothelial cells using patch-clamp electrophysiology. Temporal responses to flow were recorded using the whole-cell configuration with physiological ionic gradients and steady-state voltage of −60 mV. In a static bath, $Pkd2^{fl/fl}$ endothelial cells generated a mean steady-state inward current of ~−80 pA (*Figure 4A,C*). Flow stimulated an initial, transient peak increase in mean inward current of ~−21 pA that was followed by a sustained reduction in inward current that plateaued at ~−11 pA in $Pkd2^{fl/fl}$ cells (*Figure 4*-C). In the continuous presence of flow, the removal of bath Ca$^{2+}$ increased mean inward current to ~−53 pA in $Pkd2^{fl/fl}$ cells (*Figure 4A,C*). In a static bath, mean steady-state inward current was similar in $Pkd2^{fl/fl}$ and $Pkd2$ ecKO cells (*Figure 4A,C*). In contrast, flow activated a transient peak inward current in $Pkd2$ ecKO cells that was only ~ 15% of that in $Pkd2^{fl/fl}$ cells (*Figure 4A, B*). Similarly, the sustained flow-mediated reduction in inward current in $Pkd2$ ecKO cells was ~ 40% of that in $Pkd2^{fl/fl}$ cells (*Figure 4A,C*). In the continuous presence of flow, removal of bath Ca$^{2+}$ resulted alsoin a smaller increase in inward current in $Pkd2$ ecKO cells than in $Pkd2^{fl/fl}$ cells (*Figure 4A,C*). Specifically, under flow Ca$^{2+}$ removal increased inward current only ~ 25 pA in $Pkd2$ ecKO cells, which was ~ 48.8% of the response in $Pkd2^{fl/fl}$ cells (*Figure 4A,C*). This differential response to Ca$^{2+}$ removal in $Pkd2^{fl/fl}$ and $Pkd2$ ecKO endothelial cells was due to flow, as inward current in a static condition was similar in cells of both genotypes regardless of whether the bath solution contained Ca$^{2+}$ or was Ca$^{2+}$-free (*Figure 4C*). These data demonstrate that flow stimulates a biphasic current response that is composed of an initial transient inward current followed by a sustained Ca$^{2+}$-dependent reduction in inward current in endothelial cells. Data also indicate that PKD2 channels contribute to both of these flow-mediated phases.

## PKD2-mediated Ca$^{2+}$ influx activates SK/IK channels in endothelial cells

Attenuation of the flow-mediated sustained reduction in steady-state inward current by both PKD2 knockout and extracellular Ca$^{2+}$ removal suggests the involvement of IK and SK channels. Under flow, the co-application of apamin and Tram-34, SK and IK channel blockers respectively, increased mean inward current by ~ 23.2 pA in $Pkd2^{fl/fl}$ cells (*Figure 5A,B,D*). In contrast, the apamin/tram-34-mediated increase in inward current under flow in $Pkd2$ ecKO cells was only ~ 11.6 pA or ~50% of that in $Pkd2^{fl/fl}$ cells (*Figure 5A,B,D*). In a static bath, apamin/Tram-34 produced a far smaller and

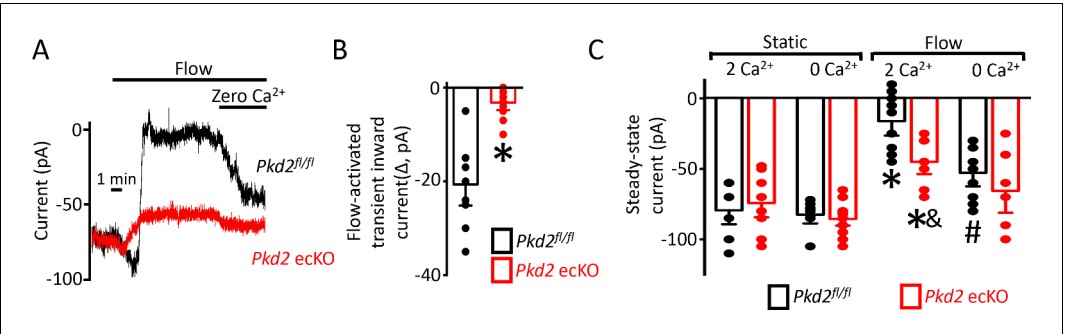

**Figure 4.** Flow reduces steady-state inward current through a PKD2-mediated, $Ca^{2+}$ influx-dependent mechanism in voltage-clamped mesenteric artery endothelial cells. (**A**) Original recordings of steady-state current modulation by flow (10 ml/min) and effect of removing bath $Ca^{2+}$ at −60 mV in endothelial cells from $Pkd2^{fl/fl}$ and $Pkd2$ ecKO mice. (**B**) Mean data for flow-induced transient inward current. n = 9 for $Pkd2^{fl/fl}$ and n = 10 for $Pkd2$ ecKO. * indicates p<0.05 versus $Pkd2^{fl/fl}$.(**C**) Mean data for steady-state currents in the presence and absence of flow and in the presence and absence of extracellular $Ca^{2+}$ ($Pkd2^{fl/fl}$: static + $Ca^{2+}$, n = 9; static with zero $Ca^{2+}$, n = 6; flow + $Ca^{2+}$, n = 9; flow with zero $Ca^{2+}$, n = 9 and $Pkd2$ ecKO: static + $Ca^{2+}$, n = 9; static with zero $Ca^{2+}$, n = 15; flow + $Ca^{2+}$, n = 8; flow with zero $Ca^{2+}$, n = 8). *p<0.05 versus static + $Ca^{2+}$ conditions in the same genotype, and indicates p<0.05 vs $Pkd2^{fl/fl}$ under the same condition, # p<0.05 versus flow + $Ca^{2+}$ in the same genotype.

similar increase in inward current in $Pkd2^{fl/fl}$ and $Pkd2$ ecKO cells (***Figure 5C,D***). These data suggest that flow stimulates PKD2-mediated $Ca^{2+}$ influx that activates SK/IK channels in endothelial cells, leading to a steady-state reduction in inward current.

Next, we tested the hypothesis that flow-stimulates vasodilation through PKD2-mediated SK/IK channel activation. Apamin/Tram-34 reduced both flow- and ACh -induced vasodilation in $Pkd2^{fl/fl}$ arteries to ~ 77% and 57% of those that occurred in the control condition (***Figure 6A,C***). Apamin/Tram-34 reduced mean ACh-induced vasodilation to ~ 54% of that in control in $Pkd2$ ecKO arteries, which was a similar reduction to that in $Pkd2^{fl/fl}$ arteries (***Figure 6A,C***). In contrast, apamin/Tram-34 did not alter flow-mediated vasodilation in $Pkd2$ ecKO arteries (***Figure 6A,C***). With static intravascular solution, bath application of apamin/Tram-34 did not alter the diameter of pressurized, myogenic $Pkd2^{fl/fl}$ or $Pkd2$ ecKO mesenteric arteries, indicating that SK and IK channels are not active in the absence of flow or ACh (***Figure 6B,C***). These data indicate that flow stimulates PKD2-mediated SK/IK channel activation in endothelial cells to induce vasodilation. In contrast, ACh stimulates vasodilation via a PKD2-independent SK/IK channel-mediated mechanism.

## PKD2 channel activation is essential for flow-mediated eNOS activation in endothelial cells

Flow stimulates nitric oxide synthase (NOS) in endothelial cells, but the significance of PKD2 channels to this activation mechanism is unclear (***Fleming, 2010***; ***Balligand et al., 2009***; ***Garcia and Sessa, 2019***). Phosphorylation of bovine eNOS at serine 1179 and human eNOS at serine 1177 leads to activation (***Fulton et al., 1999***; ***Dimmeler et al., 1999***). Western blotting was performed to measure both eNOS protein phosphorylated at serine 1176 (p-eNOS (S1176)) and total eNOS protein in mouse mesenteric arteries. Intravascular flow (15 dyn/cm², 5 min, 37°C) increased mean p-eNOS (S1176) protein ~ 1.4 fold in $Pkd2^{fl/fl}$ arteries, but only ~ 1.08 fold in $Pkd2$ ecKO arteries (***Figure 7A, B***). In contrast, flow did not alter total eNOS in either genotype (***Figure 7A,B***). L-NNA, a NOS inhibitor, reduced flow-mediated vasodilation to ~ 64% of control in pressurized $Pkd2^{fl/fl}$ arteries and to ~ 83% of control in $Pkd2$ ecKO arteries (***Figure 7C,D***). Thus, the L-NNA-induced reduction in flow-mediated vasodilation in $Pkd2$ ecKO arteries was ~ 47% of that in $Pkd2^{fl/fl}$ arteries (***Figure 7C, D***). These data indicate that PKD2 channels are key for flow to activate eNOS in endothelial cells and to elicit vasodilation through this mechanism in mesenteric arteries.

## $Pkd2$ ecKO mice are hypertensive

In vitro evidence that endothelial cell PKD2 channels contribute to flow-mediated vasodilation suggests that these proteins may regulate blood pressure. Telemetry measurements were performed

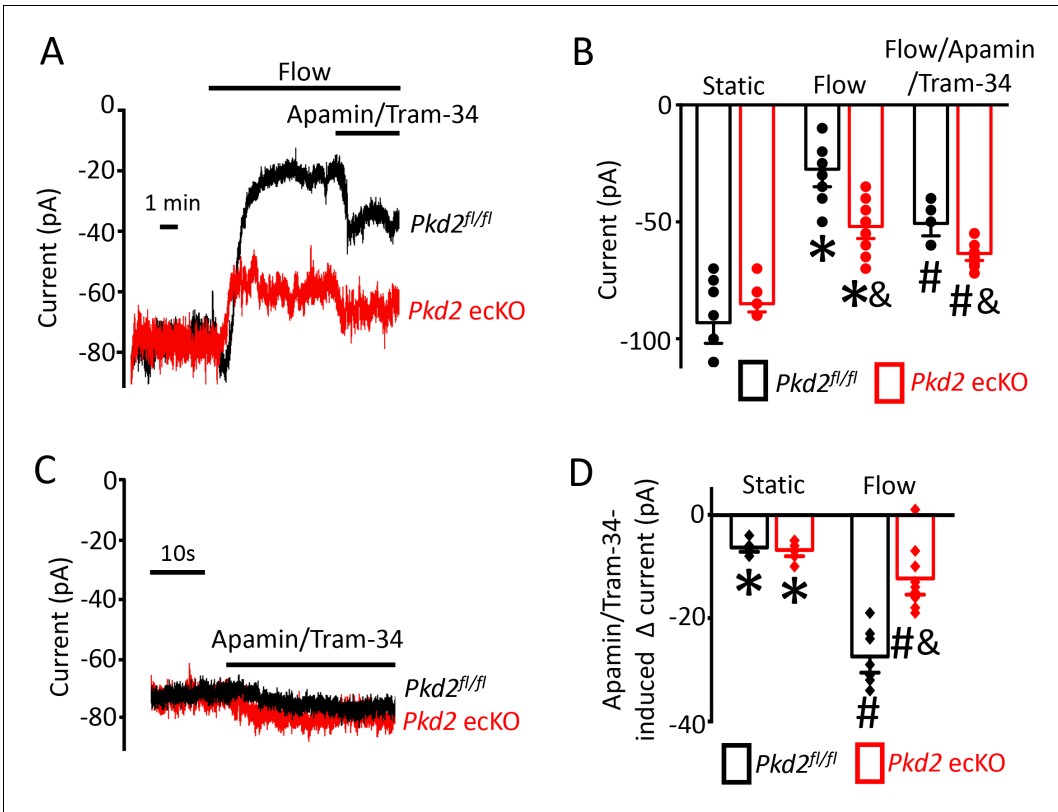

**Figure 5.** Flow-mediated PKD2 channel activation stimulates SK/IK channels in mesenteric artery endothelial cells, leading to vasodilation. (**A**) Original recordings of steady-state current modulation by flow (10 ml/min) and flow plus apamin/Tram-34 (300 nM of each) at −60 mV in mesenteric artery ECs from *Pkd2*[fl/fl] and *Pkd2* ecKO mice. (**B**) Mean data (*Pkd2*[fl/fl]: static, n = 8; flow, n = 8; flow + apamin/Tram-34, n = 7. *Pkd2* ecKO: static, n = 9, flow, n = 10; flow + apamin/Tram-34, n = 9). * indicates p<0.05 versus static in the same genotype and p<0.05 vs *Pkd2*[fl/fl] in the same conditions. # p<0.05 versus flow in the same genotype. (**C**) Original recordings of steady-state current modulation by apamin/Tram-34 (300 nM of each) in the absence of flow at −60 mV in ECs from *Pkd2*[fl/fl] and *Pkd2* ecKO mice. (**D**) Mean data comparing responses to apamin/Tram-34 in static and flow conditions at −60 mV (*Pkd2*[fl/fl]: static, n = 6; flow, n = 7. *Pkd2* ecKO: static, n = 6; flow, n = 9). *p<0.05 versus static control. # p<0.05 versus static + apamin/Tram-34 in the same genotype. and indicates p<0.05 for *Pkd2* ecKO vs *Pkd2*[fl/fl] in the same condition.

using implanted probes to measure systemic blood pressure in *Pkd2*[fl/fl] and *Pkd2* ecKO mice. Diastolic and systolic blood pressures were ~ 9 and 14 mmHg higher, respectively, in *Pkd2* ecKO than *Pkd2*[fl/fl] mice, which translated to a mean arterial pressure (MAP) that was raised by ~ 11% (*Figure 8A,B*). Locomotion was similar between genotypes, indicating that the higher blood pressure in *Pkd2* ecKO mice was not due to higher activity (*Figure 8—figure supplement 1*). Echocardiography measurements indicated that cardiac output, fractional shortening, ejection fraction and heart rate were all similar in *Pkd2*[fl/fl] and *Pkd2* ecKO mice (*Figure 8C–F*). Proximal tubule diameter and glomerular area were also similar in kidneys of *Pkd2*[fl/fl] and *Pkd2* ecKO mice, indicating no renal dysfunction (*Figure 8G–I*). These results demonstrate that flow stimulates PKD2 channels in endothelial cells to induce vasodilation and reduce systemic blood pressure.

## Discussion

Here, we investigated mechanisms of regulation and physiological functions of PKD2 channels in endothelial cells by using an inducible, conditional knockout mouse model. Endothelial cell PKD2 knockout robustly inhibits flow-mediated vasodilation, but does not alter dilation to ACh, in resistance-size arteries, suggesting stimulus-specific signaling and function. Flow stimulates PKD2 channels, leading to both Ca$^{2+}$ influx-dependent SK/IK channel activation and eNOS phosphorylation

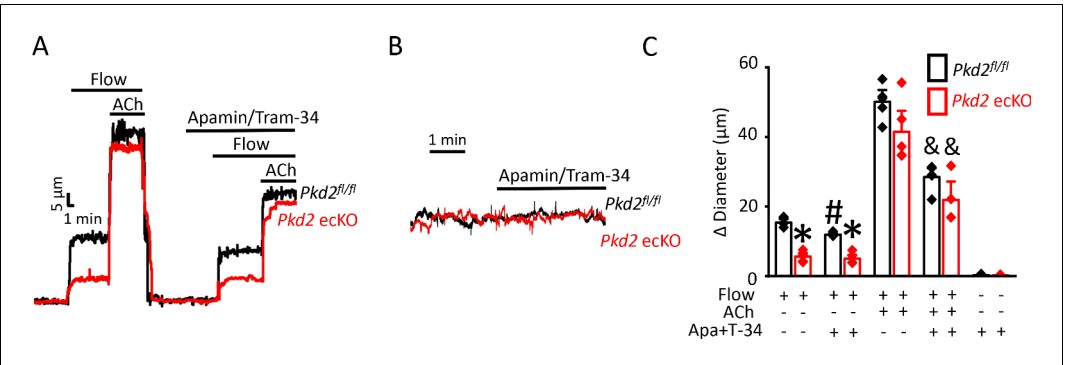

**Figure 6.** PKD2 channels contribute to intravascular flow-mediated SK/IK channel activation and vasodilation. (**A**) Representative traces illustrating responses to flow (15 dyn/cm$^2$) and flow (15 dyn/cm$^2$) + ACh (10 µM) in the presence and absence of apamin/Tram-34 (300 nM of each) in pressurized (80 mmHg) mesenteric arteries from *Pkd2*$^{fl/fl}$ and *Pkd2* ecKO mice. (**B**) Representative traces illustrating responses to apamin/Tram-34 (300 nM of each) in the absence of intravascular flow in pressurized (80 mmHg) mesenteric arteries. (**C**) Mean data (*Pkd2*$^{fl/fl}$: flow, n = 5; flow + apamin/Tram-34, n = 5; flow + ACh (10 µM), n = 5; flow + ACh (10 µM) + apamin/Tram-34, n = 5; static + apamin/Tram-34, n = 5. *Pkd2* ecKO: flow, n = 5; flow + apamin/Tram-34, n = 5; flow + ACh (10 µM), n = 5; flow + ACh (10 µM) + apamin/Tram-34, n = 4; static + apamin/Tram-34, n = 5). * indicates p<0.05 versus *Pkd2*$^{fl/fl}$ in the same condition. # indicates p<0.05 for flow + apamin/Tram-34 versus flow in the same genotype. and indicates p<0.05 for flow + ACh versus flow + ACh + apamin/Tram-34 in the same genotype.

and activation in endothelial cells (*Figure 9*). These mechanisms induce arterial hyperpolarization and vasodilation. Endothelial cell PKD2 channel knockout increased both diastolic and systolic blood pressure in mice, without effects on cardiac function or kidney anatomy. Thus, by coupling intravascular flow to vasodilation, endothelial cell PKD2 channels reduce blood pressure.

Global knockout of a gene can lead to compensatory expression of other genes that produce contrasting and contradictory results to those expected from studies on isolated cells and tissues. Whether endothelial cell TRP channels are functional could not be determined from global TRPC6, TRPM4 and TRPV4 channel knockout mice, which generated complex findings associated with compensatory mechanisms (*Earley et al., 2009*; *Mathar et al., 2010*; *Dietrich et al., 2005*; *Nishijima et al., 2014*). Global knockout of TRPM4, which is expressed in multiple cell types, including arterial smooth muscle, increased catecholamine secretion that elevated blood pressure in mice (*Mathar et al., 2010*; *Earley et al., 2004*). TRPC6 knockout resulted in upregulation of constitutively active TRPC3 channels in arterial smooth muscle cells that caused vasoconstriction and elevated blood pressure (*Dietrich et al., 2005*). Global knockout of TRPV4 channels, which are expressed in both arterial smooth muscle cells and endothelial cells, resulted in either the same or lower blood pressure than controls (*Earley et al., 2009*; *Nishijima et al., 2014*). Here, inducible, endothelial cell-specific PKD2 knockout did not alter the expression of PKD1, Piezo1, GPR68, SK3, IK or TRPV4 channels in mesenteric arteries. Flow stimulated plasma membrane Ca$^{2+}$ influx that activated SK/IK channels, producing a steady-state reduction in inward current in *Pkd2*$^{fl/fl}$ mouse endothelial cells. PKD2 knockout reduced both the flow-mediated transient inward current and the steady-state reduction in inward current that occurred, in part, due to Ca$^{2+}$-dependent SK/IK channel activation. Flow elevated intracellular Ca$^{2+}$ concentration and activated eNOS in a Ca$^{2+}$/calmodulin–dependent manner (*Fleming, 2010*; *Michel and Vanhoutte, 2010*; *Zhou et al., 2014*). The ion channel(s) responsible for these Ca$^{2+}$-dependent signaling mechanisms were unclear. Here, we show that PKD2 channels are essential to both flow-mediated Ca$^{2+}$ influx that activates SK/IK channels and to eNOS activation. PKD2 channel properties have been debated for almost two decades, particularly their ionic permeability. Recent evidence suggests that PKD2 homotetramers are voltage-dependent, outwardly rectifying and primarily permeant to Na$^+$ and K$^+$, with low Ca$^{2+}$ permeability (*Shen et al., 2016*; *Wang et al., 2019*). PKD1 and PKD2 in a 1 to 3 ratio, respectively, can also form a heterotetrameric channel that is far more permeant to Ca$^{2+}$ than PKD2 homotetramers (*Wang et al., 2019*; *Su et al., 2018*; *Zhu et al., 2011*; *Yu et al., 2009*). Whether flow stimulates PKD2 homotetramers and/or a PKD1/PKD2 heterotetramers to generate the Ca$^{2+}$ signal that activates SK/IK channels remains to be

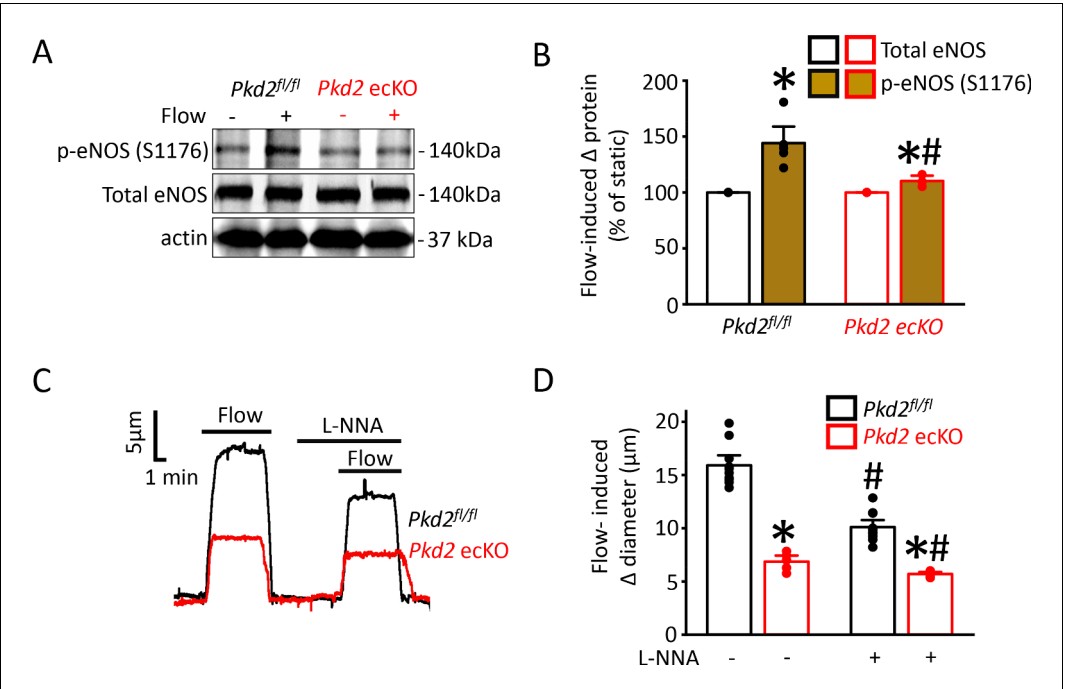

**Figure 7.** Flow-mediated PKD2 channel activation in ECs stimulates eNOS serine 1176 phosphorylation, leading to vasodilation. (**A**) Original Western blots illustrating effects of flow (15 dyn/cm$^2$) and *Pkd2* ecKO on p-eNOS (S1176) and total eNOS proteins in *Pkd2*$^{fl/fl}$ and *Pkd2* ecKO mesenteric arteries. (**B**) Mean data for flow-induced change (Δ) in proteins. n = 5 for *Pkd2*$^{fl/fl}$. n = 4 for *Pkd2* ecKO. * indicates p<0.05 versus static. # indicates p<0.05 versus same protein in *Pkd2*$^{fl/fl}$. (**C**) Representative traces demonstrating flow (15 dyn/cm$^2$)-mediated vasodilation in pressurized (80 mmHg) mesenteric arteries of *Pkd2*$^{fl/fl}$ and *Pkd2* ecKO mice in the presence and absence of L-NNA (10 μM). (**D**) Mean data. n = 10 for *Pkd2*$^{fl/fl}$. n = 5 for *Pkd2* ecKO. * indicates p<0.05 versus *Pkd2*$^{fl/fl}$ in the same condition. # indicates p<0.05 versus flow in the absence of L-NNA (10 μM) in the same genotype.

established. PKD2 has also been proposed to interact with TRPC1, TRPC3, TRPC5, TRPC7 and TRPV4 channels, but whether heterotetrameric channel formation occurs between these different proteins in native endothelial cells is poorly understood (*Miyagi et al., 2009*; *Tsiokas et al., 1999*; *Köttgen et al., 2008*; *Sutton et al., 2006*; *Du et al., 2014*; *Du et al., 2008*). Future studies should test these hypotheses.

Increasing intravascular flow produced progressive vasodilation, with the relative contribution of endothelial cell PKD2 channels to this response approximately 50%, regardless of the magnitude of shear stress. These results indicate that PKD2 channel activity is flow-dependent and show that PKD2 channels function over a broad range of shear stress to elicit vasodilation. PKD2 channels do not appear to be inherently flow-sensitive. Potential mechanisms by which flow stimulates PKD2 channels include coupling to PKD1, regulation by microtubules and/or the actin cytoskeleton, and through interaction with TRPV4 and TRPC1 (*Du et al., 2014*; *Li et al., 2006*; *Hardy and Tsiokas, 2020*; *Nauli et al., 2003*). PKD2 channel knockout did not abolish flow-mediated current modulation in endothelial cells or vasodilation in pressurized arteries, suggesting that PKD2 channel-independent mechanisms also contribute to these responses. A previous study demonstrated that flow activates Piezo1 channels in endothelial cells, leading to vasodilation and a reduction in blood pressure (*Wang et al., 2016*). In contrast, another study published that endothelial cell Piezo1 does not regulate blood pressure during inactivity, but is activated by an increase in flow during exercise, resulting in arterial depolarization and vasoconstriction (*Rode et al., 2017*). Global knockout of GPR68, a class A rhodopsin-like G protein-coupled receptor, reduced flow-mediated vasodilation in third-order mesenteric arteries, but did not alter effects of flow in first- or second-order mesenteric arteries where it is not expressed (*Xu et al., 2018*). Other proposed flow-mediated mechanisms include those via TRPV4, angiotensin type II, histamine, and bradykinin type two receptors, although recombinant expression of these proteins has also been shown to not produce flow-mediated responses

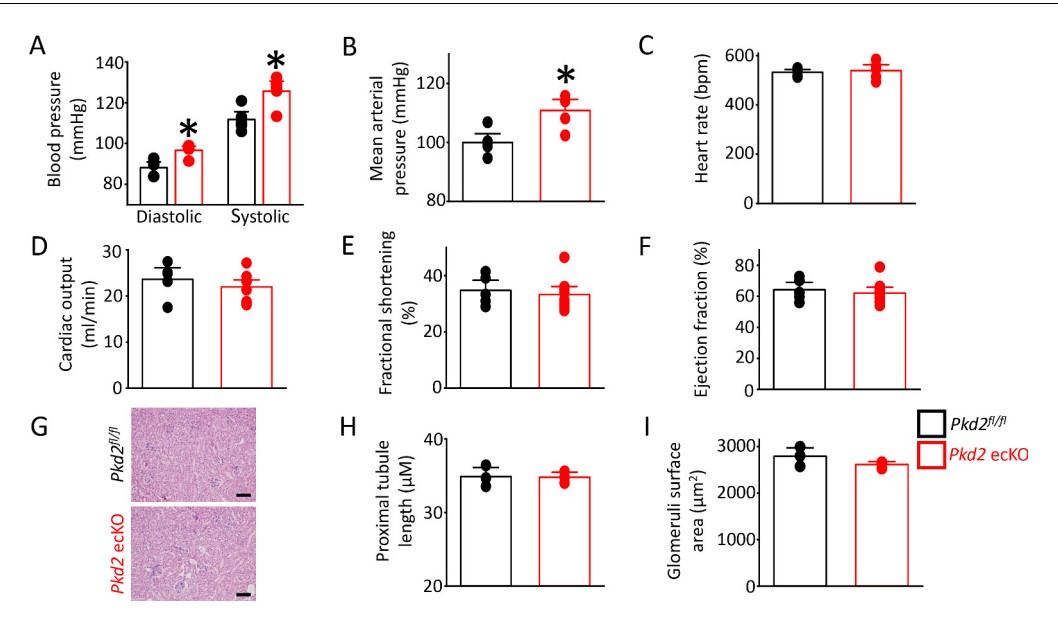

**Figure 8.** *Pkd2* ecKO elevates systemic blood pressure, but does not alter cardiac function or kidney histology. (A) Mean diastolic and systolic blood pressures in *Pkd2*^fl/fl and *Pkd2* ecKO mice (n = 5 for each). * indicates p<0.05 versus *Pkd2*^fl/fl. (B) Mean arterial blood pressure (MAP) (n = 5 for each). * indicates p<0.05 versus *Pkd2*^fl/fl. (C–F) Mean echocardiography data. Heart rate (HR), Cardiac output (CO), fractional shortening (FS) and ejection fraction (EF) (n = 5 *Pkd2*^fl/fl and n = 10 for *Pkd2* ecKO). (G) Representative images of H and E stained kidney cortex used for histological assessment. Scale bars = 100 μm. (H) Mean proximal tubule length (n = 15 proximal tubules measured for each group from three individual mice). (I) Mean glomeruli surface area (n = 75 glomeruli measured per group from three individual mice).

The online version of this article includes the following figure supplement(s) for figure 8:

**Figure supplement 1.** Locomotion is similar in *Pkd2*^fl/fl and *Pkd2* ecKO mice (n = 5 for each).

(*Zhang et al., 2009*; *Chachisvilis et al., 2006*; *Mendoza et al., 2010*; *Ramkhelawon et al., 2013*). Signaling mechanism described include the release of ATP, which binds in an autocrine manner to purinergic receptors, adrenomedullin, acting via cell surface receptors containing CALCRL, and ACh that is released by organic cation transporters and activates muscarinic receptors (*Wang et al., 2016*; *Iring et al., 2019*; *Wilson et al., 2016*). These pathways can also involve eNOS activation (*Wang et al., 2016*; *Iring et al., 2019*). Which of these mechanisms are PKD2-dependent and which are PKD2-independent remains to be established.

Endothelial cell *Pkd2* knockout attenuated flow-mediated vasodilation, but did not alter vasodilation to ACh, indicating differential signaling mechanisms. Intracellular signals and kinases that regulate PKD2 channels in endothelial cells are poorly understood. As such, it is not clear whether signaling mechanisms activated by muscarinic receptors are incapable of activating PKD2 channels. Muscarinic receptor agonists activate TRPV4 channels, leading to $Ca^{2+}$ influx, and promote endoplasmic reticulum $Ca^{2+}$ release, both of which can stimulate SK and IK channels and eNOS to produce vasodilation (*Sonkusare et al., 2012*; *Edwards et al., 2010*; *Fleming and Busse, 1999*). The relative proportion of each of these pathways to muscarinic receptor-mediated signaling differs depending on the arterial bed that is studied. Here, we show that PKD2 channel activation also stimulates IK/SK channels and eNOS. Thus IK/SK and eNOS are common downstream targets for both TRPV4 and PKD2 channels. A reasonable explanation for differential signaling elicited by flow and muscarinic receptors is that flow activates PKD2 channels in endothelial cells via a compartmentalized mechanism that excludes signaling from muscarinic receptors. Homozygous knockout of *Pkd2* is lethal in mice, precluding study of the global absence of this gene product on vascular function. ACh-induced vasodilation was attenuated due to a decrease in the availability of nitric oxide in mesenteric arteries of *Pkd2* heterozygous (*Pkd2*^+/-) mice aged between 16 and 20 weeks (*Brookes et al., 2013*). This

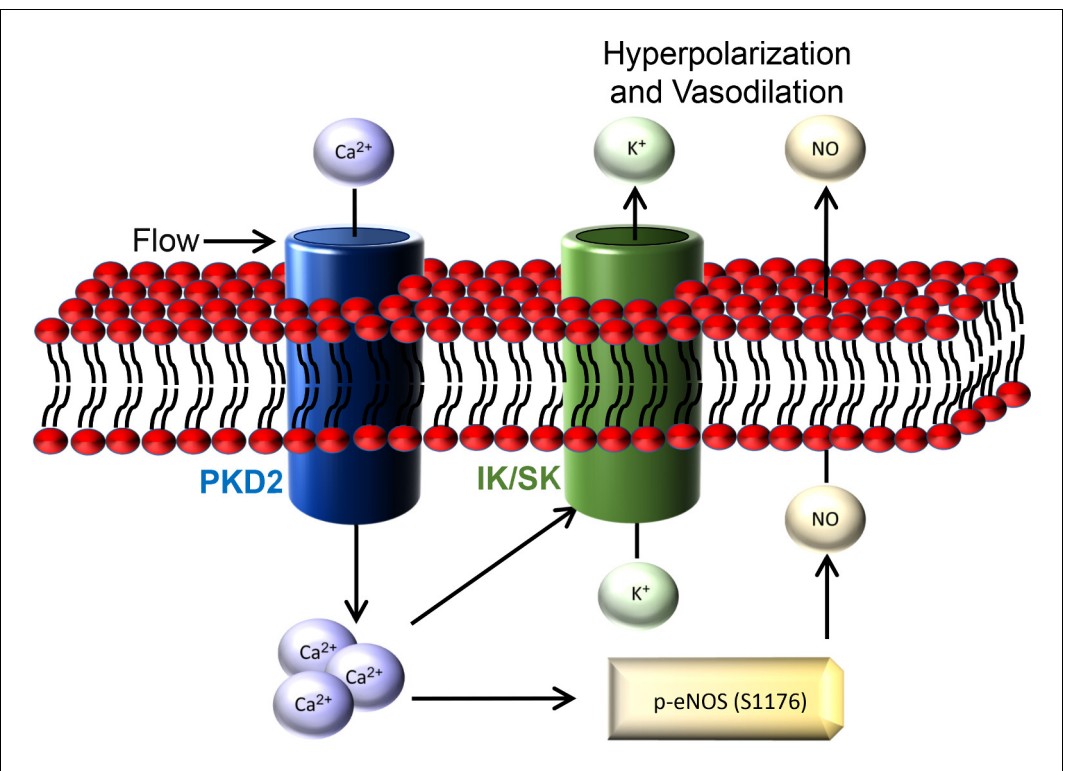

**Figure 9.** Schematic illustration of the mechanisms by which endothelial cell PKD2 channels elicit flow-mediated vasodilation.

result is in marked contrast to observations we made here where ACh-induced vasodilation was similar in mesenteric arteries of *Pkd2*^fl/fl and *Pkd2* ecKO mice. These different observations likely reflect the effects of studying short-term, endothelial cell specific PKD2 knockout versus a global and prolonged reduction in PKD2 protein that was present since gestation.

Autosomal Dominant Polycystic Kidney Disease (ADPKD) occurs due to mutations in *Pkd1* or *Pkd2* and is the most prevalent monogenic human disease worldwide, affecting 1 in 400–1000 individuals (*Torres et al., 2007*). More than 275 variants in human *Pkd2* have been identified (http://pkdb.pkdcure.org). Although ADPKD is characterized by the appearance of renal cysts, patients can develop hypertension prior to any kidney dysfunction (*Torres et al., 2007*; *Valero et al., 1999*; *Martinez-Vea et al., 2004*). Here, short-term endothelial cell PKD2 knock-out increased systemic blood pressure without inducing cardiac or renal abnormalities. These data suggest that ADPKD patients may develop hypertension due to dysfunctional endothelial cell PKD2 channels and attenuated flow-mediated vasodilation. During an increase in sustained blood flow, human ADPKD patients display loss of nitric oxide release and an associated reduction in endothelium-dependent dilation in conduit arteries, consistent with the results obtained in our mouse model (*Lorthioir et al., 2015*) As the polycystin mutation was global in these human subjects, it was not clear if the vascular deficiency was due to dysfunctional signaling in endothelial cells or another cell type that regulates endothelial cell function. Prolonged polycys-tin dysregulation in ADPKD patients may alter responses to a wide variety of other stimuli that were not studied. Future studies should investigate the effects of ADPKD-associated *Pkd2* muta-tions on endothelial cell function, arterial contractility and systemic blood pressure. Our demon-stration that endothelial cell PKD2 channels contribute to flow-mediated vasodilation and reduce blood pressure is a step forward in understanding the physiological significance of this protein and its dysfunction in patients with ADPKD and other cardiovascular diseases.

In summary, using an inducible, conditional *Pkd2* knockout mouse, we demonstrate that intravas-cular flow stimulates PKD2 channels in endothelial cells, leading to Ca²⁺-dependent SK/IK channel and eNOS activation, arterial hyperpolarization, vasodilation and a reduction in systemic blood

pressure. These results indicate that endothelial cell PKD2 channels are a major mechanistic component of functional flow-sensing in the vasculature.

# Materials and methods

## Key resources table

| Reagent type (species) or resource | Designation | Source or reference | Identifiers | Additional information |
|---|---|---|---|---|
| Strain, strain background (*M. musculus*) | Pkd2<sup>fl/fl</sup> | Baltimore PKD Core Center | PMID:20862291 | Mice with *Pkd2* gene flanked by *loxP* regions. |
| Strain, strain background (*M. musculus*) | *Cdh5*(PAC)-creERT2 | Cancer Research UK | RRID:MGI:3848984 | Mice with tamoxifen-inducible Cre recombinase that is expressed specifically in endothelial cells. |
| Strain, strain background (*M. musculus*) | Pkd2<sup>fl/fl</sup>: *Cdh5*(PAC)-creERT2 | This paper | | Mouse line created in-house by mating *Pkd2<sup>fl/fl</sup>* with *Cdh5*(PAC)-creERT2. Mice with inducible endothelial cell-specific deletion of PKD2. |
| Antibody | Anti-PKD2 (rabbit polyclonal) | Baltimore PKD Core | Rabbit mAB 3374 CT-14/4 | IF 1:200 dilution |
| Antibody | Anti-PKD2 (mouse monoclonal) | Santa Cruz | Cat# sc-47734 RRID:AB_672380 | WB 1:100 dilution |
| Antibody | Anti-PKD1 (mouse monoclonal) | Santa Cruz | Cat# sc-28331 RRID:AB_672377 | WB 1:100 dilution |
| Antibody | Anti-Piezo1 (rabbit polyclonal) | Proteintech | Cat 15939–1-AP. | WB 1:100 dilution |
| Antibody | Anti-SK3 antibody | Abcam | Cat# ab28631 RRID:AB_775888 | WB 1:100 dilution |
| Antibody | Anti-IK1 Antibody (D-5) (mouse monoclonal) | Santa Cruz | Cat# sc-365265 RRID:AB_10841432 | WB 1:100 dilution |
| Antibody | Anti-eNOS (mouse monoclonal) | Abcam | Cat# ab76198 RRID:AB_1310183 | WB 1:100 dilution |
| Antibody | Anti-p-eNOS (rabbit polyclonal) | Cell signaling Technology | Cat# 9571 RRID:AB_329837 | WB 1:100 dilution |
| Antibody | Anti-GPR68 | NOVUS Biologicals | Cat# NBP2-32747 | WB 1:100 dilution |
| Antibody | Anti-TRPV4 (clone 1B2.6) (mouse monoclonal) | Millipore Sigma | Cat# MABS466 | WB 1:100 dilution |
| Antibody | Anti-Actin (mouse monoclonal) | Millipore Sigma | Cat# MAB1501 RRID:AB_2223041 | WB 1:5000 dilution |
| Antibody | Alexa 555 secondary antibodies (anti rabbit and anti mouse) | Thermo Fisher | Cat# A-21429 (RRID:AB_141761) and # A-31570 (RRID:AB_2536180) | IF 1:400 dilution |
| Antibody | Alexa 488 secondary antibodies (anti rat) | Thermo Fisher | Cat# A-21470 RRID:AB_2535873 | IF 1:400 dilution |
| Other | Nuclear staining (DAPI) | Thermo Fisher | Cat# 3571 RRID:AB_2307445 | IF 1:1000 dilution |

## Animals

All procedures were approved by the Animal Care and Use Committee of the University of Tennessee (protocol 17–068.0). *Pkd2*$^{fl/fl}$ mice were obtained from the Baltimore PKD Core Center. *Cdh5* (PAC)-creERT2 mice were a kind gift from Cancer Research UK (*Wang et al., 2010*). *Pkd2*$^{fl/fl}$ mice with *loxP* sites flanking exons 11–13 of the *Pkd2* gene were obtained from the John Hopkins PKD Core. *Pkd2*$^{fl/fl}$ mice were crossed with tamoxifen-inducible endothelial cell-specific Cre mice (Cdh5 (PAC)-CreERT2, Cancer Research UK) to generate *PKD2*$^{fl/fl}$*:Cdh5(PAC)-CreERT2* mice. Male *Pkd2*$^{fl/fl}$*: Cdh5(PAC)-CreERT2* or *Pkd2*$^{fl/fl}$ mice (8–14 weeks of age) were injected with tamoxifen (1 mg/ml, i. p.) once per day for 5 days and studied 7–14 days after the last injection.

## Tissue preparation and endothelial cell isolation

Male mice were euthanized with isoflurane (1.5%), followed by decapitation. Mesenteric artery branches from first- to fifth-order were removed, cleaned of adventitial tissue and placed into ice-cold physiological saline solution (PSS) that contained (in mM): 112 NaCl, 6 KCl, 24 NaHCO$_3$, 1.8 CaCl$_2$, 1.2 MgSO$_4$, 1.2 KH$_2$PO$_4$ and 10 glucose, gassed with 21% O$_2$, 5% CO$_2$ and 74% N$_2$ to pH 7.4. Endothelial cells were dissociated by introducing endothelial cell basal media (Endothelial cell GM MV2, Promocell) containing 2 mg/ml collagenase type 1 (Worthington Biochemical) into the arterial lumen and left to incubate for 30–40 min at 37°C. Cells isolated from mesenteric arteries contain multiple different types that exhibit similar visual phenotypes upon enzymatic isolation. To obtain a population of endothelial cells, cell isolate was placed into endothelial cell basal media containing growth supplements (Promocell) that support only endothelial cell survival. Endothelial cells were then studied < 5 days later.

## Genomic PCR

Genomic DNA was isolated from mesenteric arteries using a Purelink Genomic DNA kit (Thermo Fisher Scientific). Reaction conditions used are outlined in the Baltimore PKD Center genotyping protocol (http://baltimorepkdcenter.org/mouse/PCR%20Protocol%20for%20Genotyping%20PKD2KO%20and%20PKD2%5Eneo.pdf). Genotyping was performed using a 3-primer strategy, with primers a (5'-CCTTTCCTCTGGTTCTGGGGAG), b (5'-GTTGATGCTTAGCAGATGATGGC) and c (5'-CTGACAGGCACCTACAGAACAGTG) used to identify floxed and deleted alleles.

## Western blotting

Mesenteric artery segments comprising second- to fifth-order vessels were used for Western blotting. For experiments examining flow-mediated regulation of NOS and p-eNOS proteins, a glass cannula was inserted into the first-order branch of a mesenteric artery segment and flow introduced through to fifth-order arteries. Proteins were separated on 7.5% SDS-polyacrylamide gels and blotted onto nitrocellulose membranes. Membranes were blocked with 5% milk and incubated with one of the following primary antibodies: Piezo1 (Proteintech), PKD1 (Santa Cruz), PKD2 (Santa Cruz), SK3 (Abcam), eNOS (Abcam), IK (Alomone), p-eNOS (Cell Signaling), GPR68 (NOVUS), TRPV4 (Millipore-Sigma) or actin (MilliporeSigma) overnight at 4°C. Membranes were washed and incubated with horseradish peroxidase-conjugated secondary antibodies at room temperature. Protein bands were imaged using a ChemiDoc Touch Imaging System (Bio-Rad), quantified using ImageJ software and normalized to actin.

## *En-face* arterial immunofluorescence

Arteries were cut longitudinally and fixed with 4% paraformaldehyde in PBS for 1 hr. Following a wash in PBS, arteries were permeabilized with 0.2% Triton X-100, blocked with 5% goat serum and incubated overnight with PKD2 primary antibody (Rabbit mAB 3374 CT-14/4: Baltimore PKD Center) at 4°C. Arteries were then incubated with Alexa Fluor 555 rabbit anti-mouse secondary antibody (1:400; Molecular Probes) and 4',6-diamidino-2-phenylindole, dihydrochloride (DAPI) (1:1000; Thermo Scientific) for 1 hr at room temperature. Arteries were washed with PBS and mounted in 80% glycerol solution. DAPI and Alexa 555 were excited at 350 nm and 555 nm with emission collected at $\leq$ 437 nm and $\geq$ 555 nm, respectively, using a Zeiss LSM 710 laser-scanning confocal microscope.

## Pressurized artery myography

Experiments were performed using isolated third- and fourth-order mesenteric arteries using PSS gassed with 21% $O_2$/5% $CO_2$/74% $N_2$ (pH 7.4). Arterial segments 1–2 mm in length were cannulated at each end in a perfusion chamber (Living Systems Instrumentation) continuously perfused with PSS and maintained at 37°C. Intravascular pressure was altered using a Servo pump model PS-200-P (Living systems) and monitored using pressure transducers. Following development of stable myogenic tone, luminal flow was introduced during experiments using a P720 peristaltic pump (Instech). Arterial diameter was measured at 1 Hz using a CCD camera attached to a Nikon TS100-F microscope and the automatic edge-detection function of IonWizard software (Ionoptix). Myogenic tone was calculated as: 100 x (1-$D_{active}$/$D_{passive}$) where $D_{active}$ is active arterial diameter and $D_{passive}$ is the diameter determined in the presence of $Ca^{2+}$-free PSS supplemented with 5 mM EGTA.

## Pressurized artery membrane potential measurements

Membrane potential was measured by inserting sharp glass microelectrodes (50–90 MΩ) filled with 3 M KCl into the adventitial side of pressurized third- and fourth-order mesenteric arteries. Membrane potential was recorded using a WPI FD223a amplifier and digitized using a MiniDigi 1A USB interface, pClamp 9.2 software (Axon Instruments) and a personal computer. Criteria for successful intracellular impalements were: (*Vane, 1994*) a sharp negative deflection in potential on insertion; (*Edwards et al., 1998*) stable voltage for at least 1 min after entry; (*Leffler et al., 2006*) a sharp positive voltage deflection on exit from the recorded cell and (*Garland et al., 2011*) a < 10% change in tip resistance after the impalement.

## Patch-clamp electrophysiology

The conventional whole-cell configuration was used to measure steady-state currents in isolated endothelial cells at a holding potential of −60 mV. The bath solution contained (in mM): NaCl 134, KCl 6, HEPES 10, $MgCl_2$ 1, $CaCl_2$ 2 and glucose 10 (pH 7.4, NaOH). $Ca^{2+}$-free bath solution was the same composition as bath solution except $Ca^{2+}$ was omitted and 1 mM EGTA added. The pipette solution contained (in mM): K aspartate 110, KCl 30, HEPES 10, glucose 10 and EGTA 1, with total $MgCl_2$ and $CaCl_2$ adjusted to give free concentrations of 1 mM and 200 nM, respectively. Free $Mg^{2+}$ and $Ca^{2+}$ were calculated using WebmaxC Standard (http://www.stanford.edu/~cpatton/webmaxcS. htm). The osmolarity of solutions was measured using a Wescor 5500 Vapor Pressure Osmometer (Logan, UT, USA). Currents were filtered at 1 kHz and digitized at 5 kHz using an Axopatch 200B amplifier and Clampex 10.4 (Molecular Devices. Offline analysis was performed using Clampfit 10.4. Flow-activated transient inward current was measured at its peak in each cell. Steady-state inward currents were calculated as the average of at least 45 s of continuous data.

## Telemetric blood pressure and locomotion measurements

Telemetric blood pressure recordings were performed by the University of Cincinnati Mouse Metabolic Phenotyping Center. Briefly, transmitters (PA-C10, Data Sciences International) were implanted subcutaneously into anesthetized mice, with the sensing electrode placed in the aorta via the left carotid artery. Blood pressures were measured prior, during and following tamoxifen injections (1 mg/ml, i.p) using a PhysioTel Digital telemetry platform (Data Sciences International). Dataquest A.R. T. software was used to acquire and analyze data.

## Echocardiography

Age- and sex-matched mice were anesthetized with isoflurane and placed on a warm pad on a recording stage of a Vevo 2100 ultrasound machine. The anterior chest was shaved and ultrasound coupling gel applied. Electrodes were connected to each limb and an electrocardiogram was recorded. Two-dimensional (short axis-guided) M-mode measurements were taken at the level of the papillary muscles using an 18–32 MHz MS400 transducer, as previously described (*Parks et al., 2016*). Images were also recorded in the parasternal long-axis. For analysis purposes, three or more beats were averaged using measurements within the same HR interval (450 ± 50 bpm) for analysis.

## Kidney histology

Kidney sections were stained with H and E and examined by Probetex, Inc (San Antonio, Texas). Briefly, image analysis was performed to measure glomerular size and tubular cross-sectional diameter. Glomerular size was measured by tracing the circumference of each of 25 random glomeruli and surface area calculated using the polygonal area tool of Image-Pro 4.5 image analysis software calibrated to a stage micrometer. Tubular size was measured using the linear length tool of Image-Pro 4.5 imaging software. The tracing tool was applied at the diameter of cross-sectional profiles of 5 proximal tubules/image (total of 25/section). Glomerular and tubular images were calibrated to a stage micrometer and data was transferred to an Excel spreadsheet and statistical analysis performed by Excel analysis pack.

## Statistical analysis

OriginLab and GraphPad InStat software were used for statistical analyses. Values are expressed as mean ± SEM. Student t-test was used for comparing paired and unpaired data from two populations and ANOVA with Holm-Sidak post hoc test used for multiple group comparisons. $p < 0.05$ was considered significant. Power analysis was performed to verify that the sample size gave a value of > 0.8 if P was > 0.05. Kidney histology, blood pressure and cardiac function experiments were all done single blind, wherein the person performing both the experiments and analysis of the results was not aware of the mouse genotype.

## Acknowledgements

This study was supported by NIH/NHLBI grants HL133256 and HL137745 to JHJ, American Heart Association (AHA) Scientist Development Grants to SB (16SDG27460007) and MDM (15SDG22680019) and AHA Postdoctoral Fellowships to C M (20POST35210200) and R H (16POST30960010).

## Additional information

### Funding

| Funder | Grant reference number | Author |
| --- | --- | --- |
| National Institutes of Health | HL133256 | Jonathan H Jaggar |
| National Institutes of Health | HL137745 | Jonathan H Jaggar |
| American Heart Association | 16SDG27460007 | Simon Bulley |
| American Heart Association | 15SDG22680019 | M Dennis Leo |
| American Heart Association | 20POST35210200 | Charles E MacKay |
| American Heart Association | 16POST30960010 | Raquibul Hasan |

The funders had no role in study design, data collection and interpretation, or the decision to submit the work for publication.

### Author contributions

Charles E MacKay, Conceptualization, Data curation, Formal analysis, Funding acquisition, Validation, Investigation, Visualization, Methodology, Writing - original draft, Writing - review and editing; M Dennis Leo, Data curation, Formal analysis, Validation, Investigation; Carlos Fernández-Peña, Wen Yin, Investigation; Raquibul Hasan, Alejandro Mata-Daboin, Jesse Gammons, Salvatore Mancarella, Data curation, Formal analysis, Validation, Investigation, Methodology; Simon Bulley, Investigation, Methodology; Jonathan H Jaggar, Conceptualization, Supervision, Funding acquisition, Validation, Writing - original draft, Project administration, Writing - review and editing

### Author ORCIDs

Charles E MacKay https://orcid.org/0000-0002-2875-0677
Carlos Fernández-Peña http://orcid.org/0000-0002-0726-3204

Simon Bulley (iD) http://orcid.org/0000-0001-5985-0489
Jonathan H Jaggar (iD) https://orcid.org/0000-0003-1505-3335

## Ethics

Animal experimentation: All procedures were approved by the Animal Care and Use Committee of the University of Tennessee (protocol 17-068.0).

## Decision letter and Author response

Decision letter https://doi.org/10.7554/eLife.56655.sa1
Author response https://doi.org/10.7554/eLife.56655.sa2

## Additional files

### Supplementary files

• Transparent reporting form

### Data availability

All data generated or analysed during this study are included in the manuscript and supporting files.

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
