## [Decision Letter]

Thank you for submitting your article "Intravascular flow stimulates PKD2 (polycystin-2) channels in endothelial cells to reduce blood pressure" for consideration by *eLife*. Your article has been reviewed by three peer reviewers, one of whom is a member of our Board of Reviewing Editors, and the evaluation has been overseen by Richard Aldrich as the Senior Editor. The reviewers have opted to remain anonymous.

The reviewers have discussed the reviews with one another and the Reviewing Editor has drafted this decision to help you prepare a revised submission.

Summary:

The manuscript by MacKay et al. uses an endothelial cell (EC)-specific polycystin-2 (*Pkd2*)-knockdown model that the authors developed to establish that the transient receptor potential family member, TRPP1, also known as PKD2, contributes to flow-induced dilation in mesenteric resistance arteries via an IK/SK channel- and NO-dependent mechanism. The authors further show that blood pressure is elevated in *Pkd2* ecKD mice in the absence of alterations in other cardiovascular metrics or kidney function, suggesting a role for this channel in regulating vasodilation in vivo under baseline conditions. The experimental approach is appropriate, the results are generally logically presented, and the conclusions follow from the results. Remaining experimental issues are relatively minor, and there are some additional concerns regarding the presentation.

Essential revisions:

1) The authors should comment on the potential mechanism by which flow increases PKD2 activity.

2) What do the authors predict the Po-shear stress relationship of PKD2 would be and how would it influence Figure 2E? How would it compare to other flow sensitive channels?

3) This lab reported PKD2 expression smooth muscle cells and that these channels contribute to the development of myogenic tone. Together with the new data, that would imply that PKD2 channels are activated by stretch and flow. Are these modes of activation synergistic? Also, how would the function of PKD2 in smooth muscle and endothelial cells would balance to produce a specific vessel response to a change in pressure and flow?

4) It would be helpful to include in Figure 2E a third data set showing the PKD2-sensitive component of flow induced changes in diameter.

5) The observation that loss of ecPKD2 channels does not alter the response of ECs to ACh. How are these channels modulated by kinases such by mediators of the muscarinic response? Perhaps PKD2 channels are not expressed at distances sufficiently close to AKAPs to be modulated by this signaling pathway.

6) The approximate ~33% reduction in PKD2 expression in EC-specific tamoxifen-treated *Pkd2^fl/fl^:Cdh5(PAC)-creERT2* mice (Western blotting) seems pretty meager. The description of the methods on this point implies that these data were obtained from whole artery segments, including the smooth muscle layer, which could account for the seemingly modest knockdown. Although immunohistochemistry results suggest a more robust knockdown, these results are qualitative for the most part. In any case, methods used to prepare samples should be clarified and possible explanations for the underwhelming knockdown effect in whole artery segments should be discussed.

7) Data shown in Figures 4 and 5, taken together with data in Figure 6, suggest that IK/SK and NO should make additive contributions to flow-induced dilation. (There is no obvious mechanism for these two pathways to act in series.) Does application of apamin/Tram-34 together with L-NNA further reduce flow-induced dilation compared with either intervention alone? These data should be included if available.

Presentation issues:

1) The authors refer to their model as an EC-specific *Pkd2* knockout. It isn't. It's an EC-specific knockdown (ecKD) model-a fact that might contribute to observed incomplete effects (in addition to the suggested involvement of PKD2-independent mechanisms). Descriptions in the text should be modified to reflect this.

2) It's not clear from Figure 4A and B (or the Materials and methods) at what point in the traces transient responses were measured. What is clear from Figure 4A is that the direction of the response in *Pkd2*-ecKO arteries at a point that would appear to constitute a transient response is opposite that in *Pkd2^fl/fl^* arteries. These data suggest that change in current for *Pkd2*-ecKO arteries is positive, whereas data in Figure 4B indicate a small negative response (about -5 pA). Please clarify this apparent discrepancy or select a more representative trace. Also, explicitly indicate in the Materials and methods the interval over which transient responses were measured.

3) The sustained, flow-induced reduction in steady state current does not, in and of itself, provide a rationale for the involvement of SK and/or IK channels. The beginnings of a rationale for this supposition, however, might be found in the fact that this reduction is attenuated in mice with EC knockdown of the (minimally?) Ca^2+^-permeable PKD2 channel, which might arguably suggest disruption of a Ca^2+^-dependent process that could involve IK/SK channels. The logical link presented in the original should be modified to make it clear.

---

## [Author Response]

Essential revisions:1) The authors should comment on the potential mechanism by which flow increases PKD2 activity.

We agree and have now included potential mechanisms in the Discussion.

2) What do the authors predict the Po-shear stress relationship of PKD2 would be and how would it influence Figure 2E? How would it compare to other flow sensitive channels?

While we understand this request, we feel it is premature to answer this question. Data shown in Figure 2E represent shear stress-mediated activation of endothelial cell PKD2 channels, which stimulates multiple downstream signaling mechanisms, leading to communication between endothelial and smooth muscle cells, resulting in vasodilation, which is the end point that is measured in this figure. It is also unclear whether shear stress directly or indirectly activates PKD2 channels. We respectfully decline this request as we consider it speculative to extrapolate shear stress promoting vasodilation to changes in PKD2 channel P_O_ elicited by shear stress.

3) This lab reported PKD2 expression smooth muscle cells and that these channels contribute to the development of myogenic tone. Together with the new data, that would imply that PKD2 channels are activated by stretch and flow. Are these modes of activation synergistic? Also, how would the function of PKD2 in smooth muscle and endothelial cells would balance to produce a specific vessel response to a change in pressure and flow?

This is an interesting point. Of note, we demonstrated that smooth muscle cell PKD2 channels do not contribute to the myogenic response in mesenteric arteries, which is the preparation we used in this paper under review at *eLife*. Thus, the equilibrium that you suggest due to regulation of PKD2 channels in both cell types would not occur in mesenteric arteries. Pressure stimulates vasoconstriction through the activation of PKD2 channels in smooth muscle cells of hindlimb arteries, but we have not shown that endothelial cell PKD2 channels contribute to flow-mediated vasodilation in this vascular bed (Bulley et al., 2018). Therefore, we consider it premature to speculate on a potential equilibrium mediated by PKD2 in both cell types. We recognize the impact of this important concept and will discuss it in a future study should results consistent with this mechanism be obtained in hindlimb arteries.

4) It would be helpful to include in Figure 2E a third data set showing the PKD2-sensitive component of flow induced changes in diameter.

We agree and have now included this dataset in Figure 2E.

5) The observation that loss of ecPKD2 channels does not alter the response of ECs to ACh. How are these channels modulated by kinases such by mediators of the muscarinic response? Perhaps PKD2 channels are not expressed at distances sufficiently close to AKAPs to be modulated by this signaling pathway.

This is an interesting hypothesis, which we have now discussed in the manuscript. Intracellular signals and kinases that regulate PKD2 channels in endothelial cells are poorly understood. As such, we are unable to conclude whether signaling mechanisms activated by muscarinic receptors are simply incapable of activating PKD2 channels. Our data here and that previously published by others do indicate that PKD2 and TRPV4 channels both activate IK/SK channels and eNOS in endothelial cells. Based on these observations, we agree that compartmentalized signaling is likely to underlie differential regulation of PKD2 channels by flow and muscarinic receptors.

6) The approximate ~33% reduction in PKD2 expression in EC-specific tamoxifen-treated Pkd2^fl/fl^:Cdh5(PAC)-creERT2 mice (Western blotting) seems pretty meager. The description of the methods on this point implies that these data were obtained from whole artery segments, including the smooth muscle layer, which could account for the seemingly modest knockdown. Although immunohistochemistry results suggest a more robust knockdown, these results are qualitative for the most part. In any case, methods used to prepare samples should be clarified and possible explanations for the underwhelming knockdown effect in whole artery segments should be discussed.

As suggested, we have expanded our explanation of these results. Western blotting was performed to quantify proteins in second- through fifth-order branches of mesenteric arteries, as stated in the Materials and methods and now in the Results. The reduction in total PKD2 protein in mesenteric arteries of *Pkd2* ecKO mice is entirely expected. Smooth muscle cells, which also express PKD2 channels, are far more abundant than endothelial cells in these mesenteric arteries. Our data here are consistent with our previous observation that smooth muscle cell-specific PKD2 knockout reduced total mesenteric arterial wall PKD2 protein by ~75 %. Thus, data are consistent with endothelial cells containing 25-30 % of PKD2 protein in the arterial wall, consistent with the Western blotting results shown in Figure 1A and B.

7) Data shown in Figures 4 and 5, taken together with data in Figure 6, suggest that IK/SK and NO should make additive contributions to flow-induced dilation. (There is no obvious mechanism for these two pathways to act in series.) Does application of apamin/Tram-34 together with L-NNA further reduce flow-induced dilation compared with either intervention alone? These data should be included if available.

We agree that this would be interesting to investigate. However, we did not perform this experiment and do not have those data to include in this revised manuscript.

Presentation issues:1) The authors refer to their model as an EC-specific Pkd2 knockout. It isn't. It's an EC-specific knockdown (ecKD) model-a fact that might contribute to observed incomplete effects (in addition to the suggested involvement of PKD2-independent mechanisms). Descriptions in the text should be modified to reflect this.

As suggested, we have expanded the text to clarify both our description and the terminology used. We agree that genetic modification may not always completely abolish protein, although there are a wide variety of explanations for such results. The inducible, conditional *Cdh5*(PAC)-creERT2 mouse model we used here to generate our *Pkd2* ecKO mouse has been used by many other groups to knockout proteins in endothelial cells. These papers referred to their mouse models as conditional “knockouts”. Rarely did earlier studies provide evidence that the targeting of a specific gene actually reduced the amount of protein in endothelial cells of mice. Even more uncommon has been for investigators to measure the amounts of other proteins to establish the specificity of knockout. Here, we used genomic PCR, Western blotting and immunofluorescence to measure *Pkd2* gene recombination, arterial PKD2 protein and endothelial cell PKD2 protein, respectively. We also measured the expression levels of eight other proteins, which did not change. As we describe in our response to your comment “Revision: #6”, the amount of reduction in PKD2 protein in arteries of the *Pkd2* ecKO mouse is expected. The remaining protein in knockout arteries corresponds to that in smooth muscle cells. The terminology used in the literature to describe the genetic approach we employed here is “knockout”. It is up for debate whether “knockout” should only be used for mouse models where it is clearly demonstrated that the protein is completely abolished. “Knockdown” is used to describe mouse models where genetic approaches have been intentionally used to partially reduce the expression of a protein, such as with heterozygotes. As we have no direct evidence to indicate that the mouse line we created is a partial knockdown, we respectfully prefer to use the common terminology of “knockout”. Using this term will maintain consistent nomenclature in the literature and reduce confusion for readers regarding the approaches we used to produce our mouse model.

2) It's not clear from Figure 4A and B (or the Materials and methods) at what point in the traces transient responses were measured. What is clear from Figure 4A is that the direction of the response in Pkd2-ecKO arteries at a point that would appear to constitute a transient response is opposite that in Pkd2^fl/fl^ arteries. These data suggest that change in current for Pkd2-ecKO arteries is positive, whereas data in Figure 4B indicate a small negative response (about -5 pA). Please clarify this apparent discrepancy or select a more representative trace. Also, explicitly indicate in the Materials and methods the interval over which transient responses were measured.

Thank you for this suggestion. We have now clarified in the Materials and methods and Results that the transient inward current was measured at its peak in each cell. We have replaced the trace for *Pkd2* ecKO in Figure 4A for a more representative one in which the flow-activated transient inward current can be clearly seen. We have now also stated that steady-state inward currents were calculated as the average of at least 45 seconds of continuous data.

3) The sustained, flow-induced reduction in steady state current does not, in and of itself, provide a rationale for the involvement of SK and/or IK channels. The beginnings of a rationale for this supposition, however, might be found in the fact that this reduction is attenuated in mice with EC knockdown of the (minimally?) Ca^2+^-permeable PKD2 channel, which might arguably suggest disruption of a Ca^2+^-dependent process that could involve IK/SK channels. The logical link presented in the original should be modified to make it clear.

As suggested, this logical link has been modified to improve clarity.